# On the Accuracy of Particle Image Velocimetry with Citizen Videos—Five Typical Case Studies

Evangelos Rozos *, Katerina Mazi and Spyridon Lykoudis

Institute for Environmental Research & Sustainable Development, National Observatory of Athens, GR 152 36 Athens, Greece; kmazi@noa.gr (K.M.); slykoud@yahoo.com (S.L.)
* Correspondence: erozos@noa.gr; Tel.: +30-210-810-9125

**Abstract:** The application of image velocimetry to measure surface streamflow velocities requires meticulous preparation, including surveying and securing both the existence of floating features on the water surface, and, as in every hydrometry method, appropriate hydraulic conditions (e.g., uniform flow, turbulent velocity profile, etc.). Though these requirements can be easily satisfied when all stages involved in image velocimetry are prepared and executed by specialists, this is not guaranteed when the video footage is recorded by citizens. This kind of spontaneously obtained data are frequently the only available information of extreme flood events; therefore, and despite their non-scientific origin and standardization, these data are very important for hydrology. In this study, we evaluate image velocimetry under a variety of conditions, including conditions resembling citizen videos. Furthermore, we conclude on the manual analysis as a means of verification of the accuracy of the velocity estimations. An interesting finding from the case study with non-uniform flow conditions was that the surface velocities occurring at the middle section of the river, estimated using large-scale particle image velocimetry algorithms, exhibited a significant error, whereas the manual estimation was more accurate. This finding calls for further investigation and a more careful approach in similar conditions.

**Keywords:** image velocimetry; uncertainty analysis; hydraulic conditions; seeding characteristics; manual video analysis; non-uniform flow; citizen data

## 1. Introduction

Image velocimetry has been proven to be a reliable method for estimating surface velocities in natural reaches and artificial channels. Though this method has been used in experimental hydraulics for decades [1], it was various recent technological advances (i.e., unmanned aerial vehicles, light and cheap high-definition cameras, modern, powerful personal computers, etc.) that created new options for field survey and allowed this method to become popular in hydrological applications [2–4]. Image velocimetry comes in many "flavors", e.g., large-scale particle image velocimetry (LSPIV) [5,6], space–time image velocimetry [7], optical tracking velocimetry [8], Kande–Lucas–Tomasi image velocimetry [9], etc. These "flavors" differ in the method employed to detect the distance traveled by floating features between two subsequent frames. From this distance, along with the known video frame rate, the velocity of the features, and hence of the water at the surface, can be estimated.

Image velocimetry has some prerequisites for being applied successfully in hydrometry. First of all, as in every stream-gauging technique, the selected site should comply with certain criteria [10] (pp. i.2-4–i.2-5). Then, in order to successfully detect features on the water surface, they must have sufficient density (frequency of occurrence in the field of view) and be uniformly distributed over the cross-section. According to a study [11], the tracers' density, dispersion, and homogeneity of shapes all have similar significance in the LSPIV accuracy, with the first being correlated negatively with the estimation error, whereas the last two are correlated positively. Other researchers have introduced

the seeding distribution index (SDI) to combine all these features into a single index that is monotonically related with the estimation errors. Selecting the range of frames that minimizes this index from a video helps image velocimetry to yield more accurate surface velocity estimations [12].

The hydraulic conditions (type of flow, bed geometry, etc.) play an equally major role in the performance of image velocimetry applications, not only as far as hydrometry standards are concerned, but also because they influence the seeding properties. For example, the seeding density will tend to be lower in higher flows but more evenly distributed. In videos recorded by non-experts displaying flows with severe hydraulic conditions [13], there are various factors that need to be considered. Boursicaud et al. [14] studied the impact of some of these factors, including camera distortion, shake or movement, resolution, etc., and suggested excluding the videos in which the flow is too wavy or non-uniform. However, being selective is not a choice in such cases where data are usually extremely rare, and effort has to be undertaken in treating these data in order to make them usable.

In this study, we employed two LSPIV tools, Fudaa-LSPIV [15] and Free-LSPIV [16], to study the influence of hydraulic conditions on the accuracy of the estimated surface velocity via LSPIV. The first tool was selected because it is a reputable software co-developed by Électricité de France (EDF), and the second, because it can provide the confidence intervals of the estimated surface velocities. The reason for employing two tools was to ensure that the derived conclusions are not specific to any individual tool, but concern the same basic concept, which is the accurate detection and interpretation of the features' displacement and its adequacy for computing the surface water velocity. These tools were applied to five case studies with varying hydraulic conditions and seeding properties to derive conclusions regarding their impact on the accuracy of the estimated velocities. The first three case studies correspond to situations in which the field survey is performed by experts, whereas the remaining two to conditions correspond to situations where videos are recorded by citizens.

## 2. Materials and Methods

### 2.1. Fudaa-LSPIV

Fudaa-LSPIV is a particle image velocimetry tool co-developed and distributed freely by EDF and Irstea with DeltaCAD since 2010. Fudaa-LSPIV is based on a cross-correlation statistical analysis to determine the displacement of features visible on water surfaces. The similarity between an interrogation window or interrogation area (IA) at frame *i* and the displaced IA at frame *i* + 1 is measured with the normalized cross correlation coefficient [17] (p. 54). Fudaa-LSPIV employs Java to provide a user-friendly interface, but the processing algorithms are written in FORTRAN in order to achieve maximum performance. The parameters of Fudaa-LSPIV are the following:

- The area size (px). This is the size in pixels of the IA side (IA is square).
- S1, S2, S3, and S4 (px). These are the maximum displacements, in pixels, along the four directions of the IA up to which the normalized correlation is calculated.
- The minimum and maximum velocity threshold (m s$^{-1}$). These values define the upper and lower limit of the range of the acceptable estimated velocity magnitudes. Non-acceptable values are filtered out.
- The minimum and maximum Vx threshold (m s$^{-1}$). These values define the upper and lower limit of the range of the acceptable x-components of the estimated velocities.
- The minimum and maximum Vy threshold (m s$^{-1}$). These values define the upper and lower limit of the range of the acceptable y-components of the estimated velocities.
- The minimum and maximum correlation. These values define the upper and lower limit of the normalized correlation values for a displacement to be considered acceptable.

Regarding the values of the parameters, usually, a preliminary study indicates a suitable set. However, it is evident that this procedure introduces subjectivity. This is another advantage of Monte Carlo simulations, which reduce this subjectivity and estimate the influence of the uncertainty of the parameters on the results.

*2.2. Free-LSPIV*

Free-LSPIV is an open-source particle image velocimetry tool developed by the Institute for Environmental Research & Sustainable Development, National Observatory of Athens, Greece. Free-LSPIV uses the fast normalized cross-correlation algorithm [18] instead of the traditional normalized correlation. The former algorithm uses Fast Fourier Transformations to calculate the involved convolutions and is up to $10\times$ faster than the traditional one [18]. Free-LSPIV is developed in MATLAB code (compatible with GNU Octave), whereas the CPU-intensive algorithms (the fast normalized cross-correlation) are developed in C++ [19]. The parameters of Free-LSPIV are the following:

- The IA size (px $\times$ px). These are the sizes in pixels of the IA sides (IA is rectangle in Free-LSPIV, whereas it is square in Fudaa-LSPIV).
- The SA size (px $\times$ px). These are the sizes in pixels of the search area sides. SA defines the boundary within which the IA is displaced at each subsequent frame to calculate the correlation coefficient. The maximum correlation coefficient obtained corresponds to the most probable features' displacement between the sequential frames. This is equivalent to S1, S2, S3, and S4 of Fudaa-LSPIV.
- The maximum velocity threshold (m s$^{-1}$), the same parameter used in Fudaa-LSPIV.
- The minimum and maximum Vx threshold (m s$^{-1}$), the same parameters used in Fudaa-LSPIV.
- The minimum and maximum Vy threshold (m s$^{-1}$), the same parameters used in Fudaa-LSPIV.
- The minimum correlation coefficient, the same parameter used in Fudaa-LSPIV.
- The contrast adjustment parameter. This parameter controls the intensity capping [20], which is applied on the IA and SA before calculating the correlation. This is an additional feature of Free-LSPIV.

The speed of the fast normalized cross-correlation algorithm enables multiple runs of Free-LSPIV to be performed for Monte Carlo simulations. The advantages of employing Monte Carlo simulations in image velocimetry have been demonstrated by Rozos et al. [19,21]. In those studies, a pseudo-random sampling approach that combines the empirical with the triangular distribution was employed. This approach increases the efficiency (better exploration of the space) of small samples. The median value obtained using the Monte Carlo simulations is used as the best estimation of the surface velocity in this study, and the 5th and 95th percentiles as the boundaries for the 90% confidence interval of that estimation.

*2.3. Manual Method*

In this study, we also employed a manual method to estimate the surface velocities. We suggest that the proximity of the manually estimated values to the values obtained with image velocimetry allows more confidence on the estimated surface velocities. The manual estimations can be based exclusively on distinct solid floating features (e.g., leaves), the movement of which is expected to provide good information regarding the flow properties. The procedure to obtain manually the velocities is simple:

- Obtain the row and column of the feature in the initial frame $i$ (let us say $r_i$ and $c_i$) and the subsequent frame $i + 1$ (let us say $r_{i+1}$ and $c_{i+1}$). Figure 1 demonstrates how this is accomplished. According to this figure, the row and column pairs in the initial and subsequent frames are (229, 589) and (234, 589), respectively.
- Transform the row and column pairs to $(x, y)$ pairs, employing a projective transformation [22], i.e., $T(r_i, c_i) \rightarrow (x_i, y_i)$ and $T(r_{i+1}, c_{i+1}) \rightarrow (x_{i+1}, y_{i+1})$.
- If $\Delta t$ is the time interval between frames $i$ and $i + 1$, then the surface velocity can be estimated with the following formula:

$$v = \|(x_{i+1}, y_{i+1}) - (x_i, y_i)\| / \Delta t, \tag{1}$$

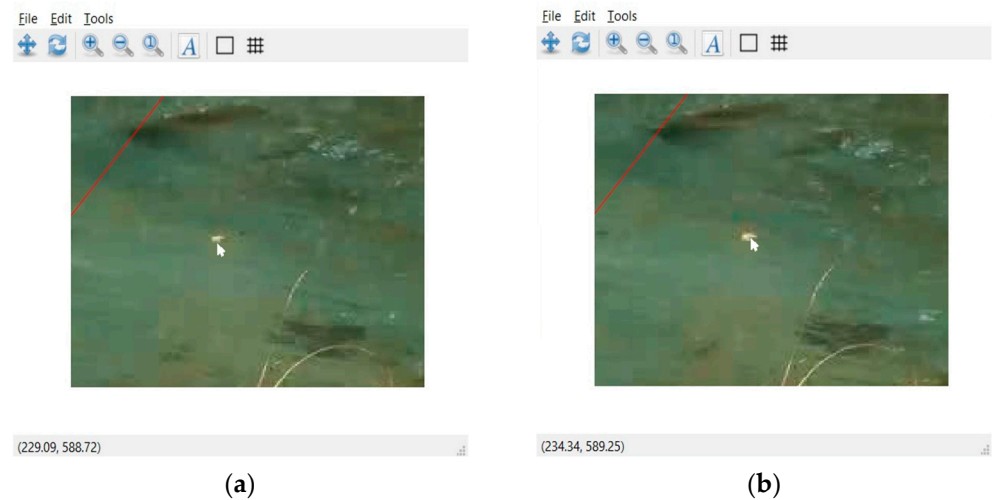

(a)  (b)

**Figure 1.** Manual estimation of the surface velocity in Loussios at Atsicholos Bridge. The cursor is placed over the feature (exactly the same point of a leaf), and the row and column pair is obtained: (**a**) frame 200 of the video; (**b**) frame 202 of the video. The frames were processed with GNU Octave.

*2.4. SVR Measurements*

Another method, also utilizing the movement of the water surface features (mainly ripples, small water waves), is Doppler radar velocimetry. An electromagnetic beam is emitted towards the water surface and backscattered to the emitting device by features on the water surface. The backscattered beam undergoes a frequency shift (Doppler effect), and this allows one to calculate the features' velocity. Usually, there is an angular spreading of the emitted beam, resulting in an ellipse-shaped measured area [23].

In this study, multiple measurements were performed with the handheld surface velocity radar (SVR) Viatronics SVR-3 Pro. This device operates at a frequency of 34.7 Ghz (Ka-Band), the beam width is 12°, the velocity range is 0.2–80 m s$^{-1}$, and the nominal measurement accuracy is ±0.3%. During measurements, the unit compensates for the pitch-down (vertical, obtained by the internal tilt sensor) and yaw (horizontal, provided by the user) angles. The maximum allowable pitch-down and yaw angles are 45° and 60°, respectively. The yaw angle is calculated using the formula $\varphi = \mathrm{atan}(d_\varphi / L)$ (see Figure 2).

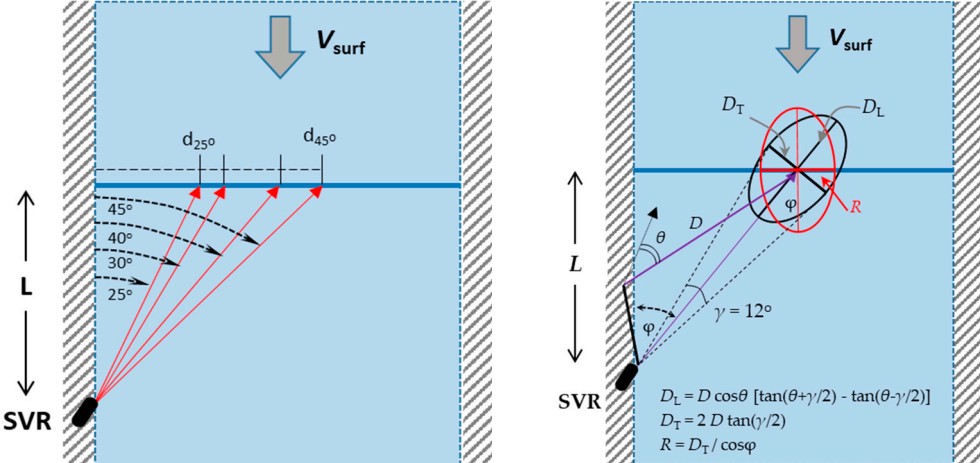

**Figure 2.** Geometry of measurements: $V_{\mathrm{surf}}$ is the surface velocity; $\theta$ is the pitch-down angle; $\varphi$ is the yaw angle; $\gamma$ is the nominal horizontal beam width; $D$ is the distance between the radar and the water surface in the line of-sight direction; $D_{\mathrm{L}}$ the longitudinal and $D_{\mathrm{T}}$ the transversal diameters of the ellipse conic; $R$ is the beam footprint on the cross-section; $L$ is the distance of the SVR from the

cross section, $d_\varphi$ is the distance between the centroid of the beam footprint on the cross-section and the bank ($d_{45°} = L$).

*2.5. Case Studies*

The following sections provide details of the case studies. The first three case studies were obtained at locations where the conditions were ideal for image velocimetry. The remaining two case studies (especially the last one) approximate conditions typical in citizen footage.

The parameters of the two employed image velocimetry tools, Fudaa-LSPIV and Free-LSPIV, are displayed in Tables A1 and A2, respectively.

2.5.1. Case Study—Kolubara River

Kolubara River is located in Central Serbia. The video and measurements were obtained from the study of Pearce et al. [24]. On the day the video was recorded, the mean flow velocity was 0.14 m s$^{-1}$, the river width was 21 m, and the maximum depth was 1.9 m. The video was recorded from an unmanned aerial vehicle (UAV) hovering 26 m above the river surface. Stabilization and orthorectification was performed with KLT-IV [25] using 6 Ground Control Points (GCPs) as references. The frame rate of the video was 24 fps, but because of the very low flow velocity, the video was sub-sampled (skipped 6 frames for every processed frame) to 4 fps. The video was recorded at a resolution of 3696 × 4994. Flow velocity measurements were obtained with the acoustic Doppler current profilers (ADCPs) SonTek RiverSurveyor M9. The reference values of the surface velocities were obtained from the ADCP data using a second-degree polynomial fit.

Figure 3 displays a frame (the part around the measured cross-section) from the Kolubara River video. Fine artificial seeding (manifesting in this figure mainly as cloudy patterns to the right of the middle of the cross-section) was distributed uniformly across the river surface.



**Figure 3.** A part of the frame from Kolubara River video; the surface velocities are obtained along the displayed red line.

2.5.2. Case Study—Murg River

Murg River is located in northeast Switzerland. The video and measurements used in this study were obtained from the study of Perks et al. [26]. On the day the video was recorded, the mean surface flow velocity was 0.80 m s$^{-1}$, the river width was 12 m, and the maximum depth was 0.35 m. The video was recorded from a UAV hovering 30 m above the river surface. Orthorectification and stabilization was performed with Photoscan (Agisoft) [27] using 14 GCPs as references. The video was recorded at a resolution of 4096 × 2160 with a frame rate of 12 fps. The flow velocity measurements were obtained with the Teledyne RDI StreamPro ADCP. The reference values of surface velocities were obtained with a complex log-fit (values '$U_{z=0}$' in Figure 12 of Detert et al. [28]) from the ADCP measurements obtained at a depth of 0.14 m (the minimum water depth ADCP can obtain measurements).

Figure 4 displays a frame (the part around the measured cross-section) from the Murg River video. Spruce wood chips were spread across the water surface, shown as the easily distinguishable white spots in Figure 4.

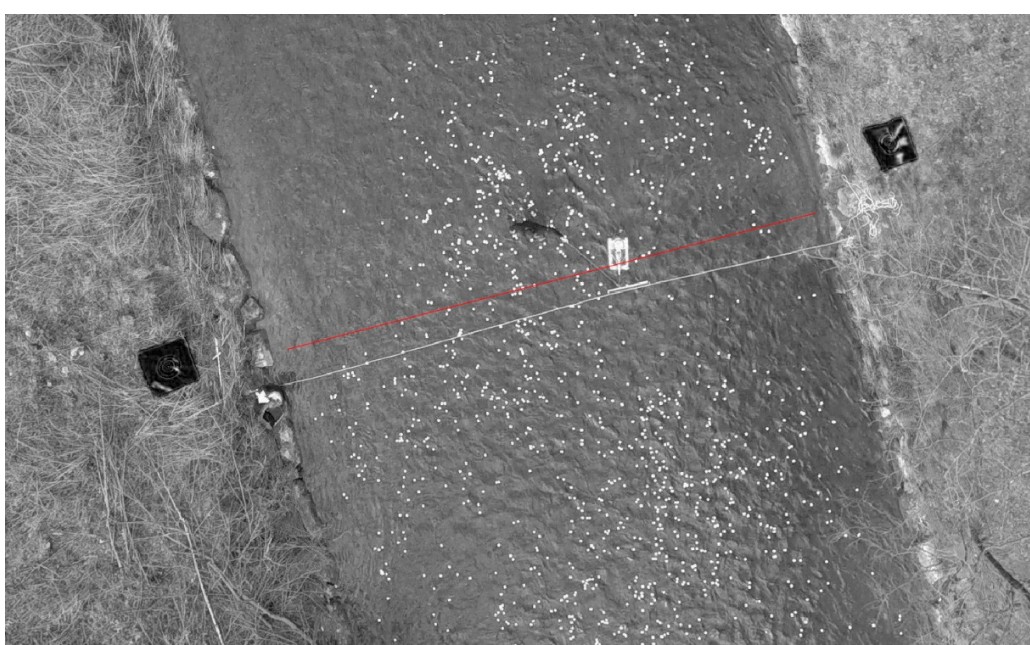

**Figure 4.** A part of a frame from Murg River video; the surface velocities are estimated along the displayed red line.

### 2.5.3. Case Study—Salmon River

Salmon River is located in British Columbia, Canada. The video and measurements used in this study were obtained from the study of Perks et al. [26]. On the day the video was recorded, the mean flow velocity was 0.58 m s$^{-1}$, the river width was 59 m, and the average depth was 0.67 m. The video was recorded from a UAV hovering 102 m above the river surface. Orthorectification and stabilization was performed with KLT-IV. The video was recorded at a resolution of 1920 × 1080 with a frame rate of 24 fps, but subsampling to a frame rate of 6 fps was employed. The measurements were made using a FlowTracker handheld acoustic Doppler velocimeter (ADV) at 60% of the depth only. These measurements were divided by 0.8 to obtain the reference values of the surface velocities (the data set provided by Perks et al. [26] includes a comment for the choice of this coefficient value citing a study of Hauet et al. [27]).

Figure 5 displays a frame (the part around the measured cross-section) from the Salmon River video. The water of the river is very clear, which makes various features on the river bottom visible. These static features do not contribute to the image velocimetry. Ripples on the water surface, which are colored red in Figure 5, are the only features moving along the flow. It is obvious that these features are denser at the middle of the section. There are moving features that correspond to the trees at the right bank, which introduce noise to the image velocimetry.

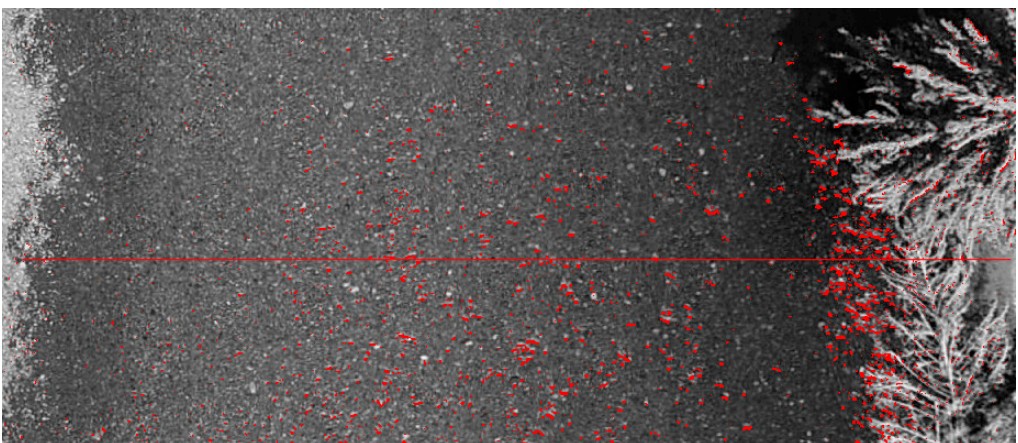

**Figure 5.** A part of a frame from the Salmon River video; the surface velocities are estimated along the displayed red line. The flowing features (ripples) are colored red.

2.5.4. Case Study—Loussios River at Atsicholos Bridge

Loussios River is located in Peloponnese, Greece. At Atsicholos Bridge, the National Observatory of Athens (NOA) has installed and operates a monitoring station (https://hydronet.noa.gr, (accessed on 19 April 2022). HYDRO-NET monitoring network, [29]). Loussios flows in a largely pristine basin, and the catchment area upstream of the station has an area of 170 km$^2$, the length of its main watercourse is 28 km, and the average basin altitude is 1081 m. A very sparse flow database exists for this river from measurements carried out by the NOA team [30]. Field survey works were performed on the 22 October 2021 at two locations with a distance of a few dozens of meters between them. The first cross-section is located at Atsicholos Bridge. At this location, the bed material is coarse sand mixed with fine gravel (1–2 cm) close to the left bank (looking upstream), large smooth pebbles (2–5 cm) at the middle of the cross-section, and then smaller and larger smooth stones (10–20 cm). On that day, the maximum flow velocity was measured to be 1.76 m s$^{-1}$, the river width was 10.5 m, and the average depth was 0.52 m. The video was recorded with a camera on a tripod on the left bank (looking upstream). As far as Free-LSPIV is concerned, the orthorectification of the raw video was accomplished with projective transformation [22] based on 4 GCPs (3 more GCPs were used for the estimation of the transformation error, which was found to range between 0.33 and 0.58 m). As far as Fudaa-LSPIV is concerned, the orthorectification was performed with its internal tool employing all 7 GCPs (see Table A3). No stabilization was required. The video was recorded at a resolution of 1280 × 720 with a frame rate of 30 fps. The measurements (the reference values) were made using a Redback Vertical Axis Type AA (operating velocity range 0.025 to 8 m s$^{-1}$, overall accuracy ±1%) ~3.5 cm below the water surface.

Figure 6 displays a frame (the part around the measured cross-section) from the Loussios at Atsicholos Bridge video. The water is clear without foam, with some ripples, and occasionally flowing leaves and other floating debris.

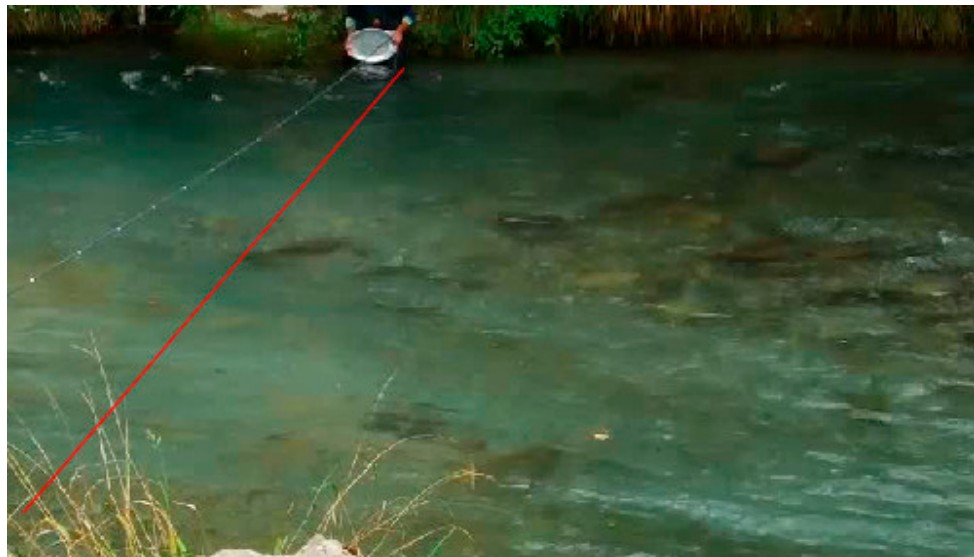

**Figure 6.** A part of the frame from the Loussios River at Atsicholos Bridge video; the surface velocities are estimated along the displayed red line.

### 2.5.5. Case Study—Loussios River Upstream Atsicholos Bridge

At this location, the riverbed is very irregular with rocks up to 3.5 m from the left bank (looking upstream), then stones and pebbles (10–15 cm) for the next 3.5 m, and large stones and sand for the remaining cross-section. The maximum flow velocity on the 22 October 2021 was measured to be 4.07 m s$^{-1}$, the river width was 9.5 m, and the average depth was 0.43 m. The video was recorded with a camera on a tripod on the left bank (looking upstream). Regarding Free-LSPIV, the orthorectification of the raw video was accomplished with projective transformation [22] based on 4 GCPs (4 more GCPs were used for the estimation of the transformation error, which was found to range between 0.09 and 0.25). As far as Fudaa-LSPIV is concerned, the orthorectification was performed with its internal tool employing all 8 GCPs (see Table A4 in Appendix B). No stabilization was required. The video was recorded at a resolution of 1280 × 720 with a frame rate of 30 fps.

The measurements (the reference values) were made using a Redback Vertical Axis Type AA (operating velocity range 0.025 to 8 m s$^{-1}$, overall accuracy ±1%) ~3.5 cm below the water surface. Measurements were also performed with the Viatronics SVR-3 Pro deployed at the left river bank, looking upstream, 6 m from the cross section. Velocity measurements were taken at four yaw angles (between the SVR's line-of-sight and flow direction) aiming at the middle section of the river, between 3 and 6 m from the left bank. The pitch-down angles $\theta$ were 20°, 18°, 18°, and 18°, and yaw angles $\varphi$ were 25°, 30°, 40°, and 45°, respectively.

Figure 7 displays a frame (the part around the measured cross-section) from the Loussios River upstream Atsicholos Bridge video, which exhibits a very disturbed water surface. A hydraulic jump probably exists at this location (the Froude number along the 3 m long middle segment of the cross-section ranges from 0.7 to 2.0).

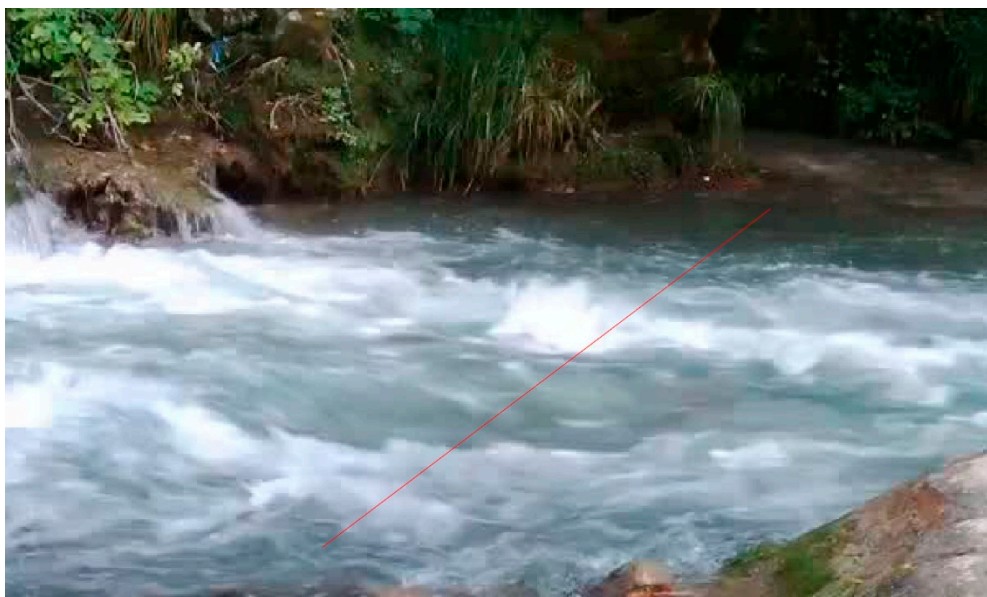

**Figure 7.** A part of the frame from the Loussios River upstream Atsicholos Bridge video; the surface velocities are estimated along the displayed red line.

## 3. Results

This section provides the estimated surface velocities via the two image velocimetry methods, the manual method, and the comparison against the reference values.

### 3.1. Kolubara River

The results of image velocimetry in the case study of Kolubara River are displayed in Figure 8. According to this figure, all methods have significant errors at the left bank. The measured velocity values are outside the Free-LSPIV 90% confidence interval at this location, but also at the middle of the cross-section. The manually estimated surface velocities are mostly outside the confidence interval. The distance from the confidence interval is higher at the cross-section locations where the velocity is lower. At these locations, the features' displacement in the two employed frames (frames 100 and 110) are less than 3 pixels, which is within the order of magnitude of the targeting accuracy of the manual method. The RMSE of Free-LSPIV, Fudaa-LSPIV, and the manual method are 0.028, 0.027, and 0.034 m s$^{-1}$, respectively.

The deviations of the image velocimetry estimations from the measurements were noticed and commented on by Pearce et al. in their study [24]. They speculated that one reason is the stabilization error, which, though very low (1.27 px) corresponding to an average velocity error of 0.024 m s$^{-1}$, is significant compared to the mean flow velocity (17% of the mean flow velocity). The other reason is the accuracy of the ADCP measurements, which are known to be characterized by increased uncertainty at low flow conditions. According to a USGS report [31] (its Figure 2) a flow velocity as low as 0.14 m s$^{-1}$, as in the Kolubara River, could result in an absolute ADCP error of 10%.

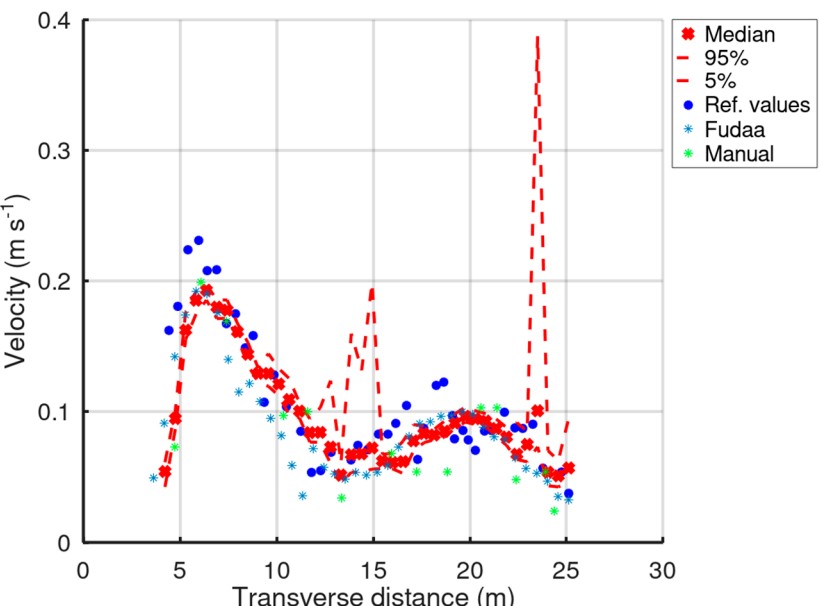

**Figure 8.** Kolubara River. The Free-LSPIV 90% confidence interval and median values, the reference values, the surface velocities obtained with Fudaa-LSPIV, and the surface velocities estimated manually.

### 3.2. Murg River

The results of image velocimetry in the case study of Murg are displayed in Figure 9. The RMSE of Free-LSPIV, Fudaa-LSPIV, and the manual method are 0.062, 0.155, and 0.119 m s$^{-1}$, respectively. As it is apparent, the results of all methods coincide, except at the locations close to the river banks, where Free-LSPIV overestimates the velocities and manifests increased uncertainty (Rozos et al. [21] suggest a method to handle this kind of uncertainty). The confidence interval is very narrow, which means that Free-LSPIV "feels" very confident about its results, and this "confidence" is verified by the proximity of the estimations to the reference values. The manual estimations at the middle section lay very close (except of one point) to the Free-LSPIV 90% confidence interval.

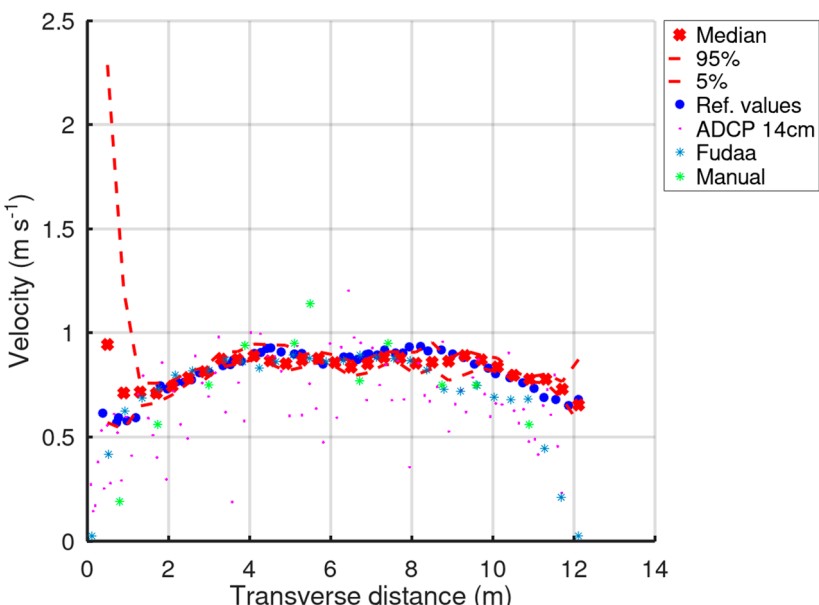

**Figure 9.** Murg River. The Free-LSPIV 90% confidence interval and median values, the reference values, the surface velocities obtained with Fudaa-LSPIV, and the surface velocities estimated manually. The observations obtained with ADCP at 14 cm are also displayed with magenta dots.

### 3.3. Salmon River

The results of image velocimetry in the case study of Salmon River are displayed in Figure 10. As shown in Figure 10, the results of all methods are very close to the reference values at the middle of the cross-section. All manual estimations are within the Free-LSPIV 90% confidence interval. The Fudaa-LSPIV and measured velocities are mostly within this confidence interval. The uncertainty, and the error, of the estimations is significant at the river banks (a method to resolve this issue is suggested in [21]). This can be attributed to the decreased density of features at these locations (see Figure 5) and the swinging objects irrelevant to the water flow (tree leaves at the right bank). The RMSE of Free-LSPIV, Fudaa-LSPIV, and the manual method are 0.277, 0.271, and 0.149 m s$^{-1}$, respectively.

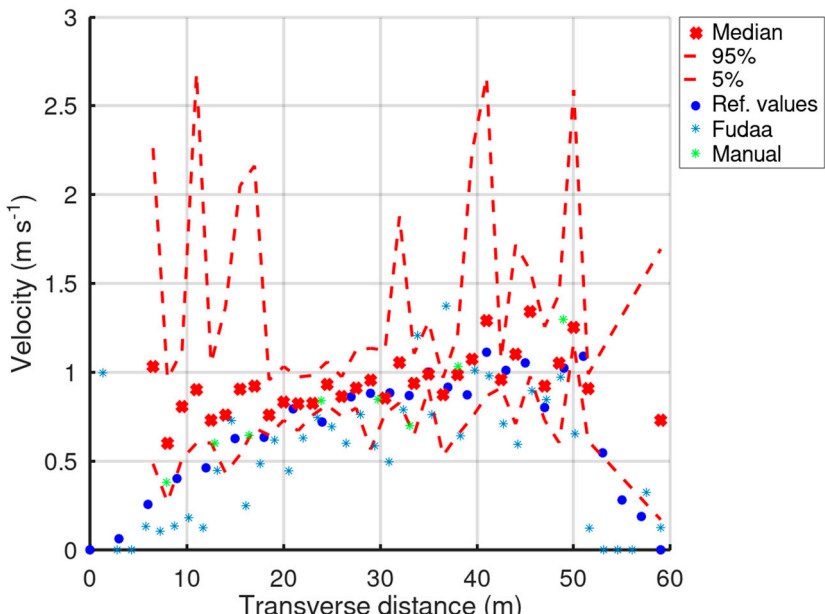

**Figure 10.** Salmon River. Free-LSPIV 90% confidence interval and median values, the reference values, the surface velocities obtained with Fudaa-LSPIV, and the surface velocities estimated manually.

### 3.4. Loussios River at Atsicholos Bridge

The results of image velocimetry in the case study of Loussios River at Atsicholos Bridge are displayed in Figure 11. In this figure, the image velocimetry uncertainty is similar at all locations of this cross-section. The estimations of all methods and the reference values are mostly within the 90% confidence interval produced by Free-LSPIV. The RMSE of Free-LSPIV, Fudaa-LSPIV, and the manual method are 0.221, 0.226, and 0.213 m s$^{-1}$, respectively.

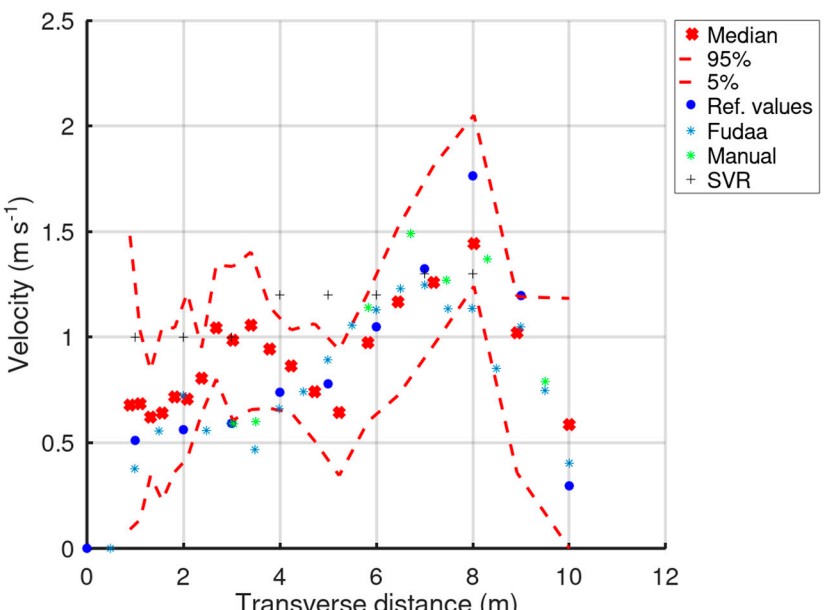

**Figure 11.** Loussios River at Atsicholos Bridge. Free-LSPIV 90% confidence interval and median values, the reference values, the surface velocities obtained with Fudaa-LSPIV, manually, and the SVR measurements.

*3.5. Loussios River Upstream*

The results of image velocimetry in the case study of Loussios River upstream Atsicholos Bridge are displayed in Figure 12. In this figure, the Free-LSPIV 90% confidence interval is narrow at most parts of the cross-section, and the reference values, the results of Fudaa-LSPIV, and the manual estimations lay mostly outside of them. The relative error of the maximum velocity is 40% for both Free-LSPIV and Fudaa-LSPIV, and 20% for the manual estimation. The corresponding RMSEs are 0.964, 1.425, and 0.907 m s$^{-1}$, respectively.

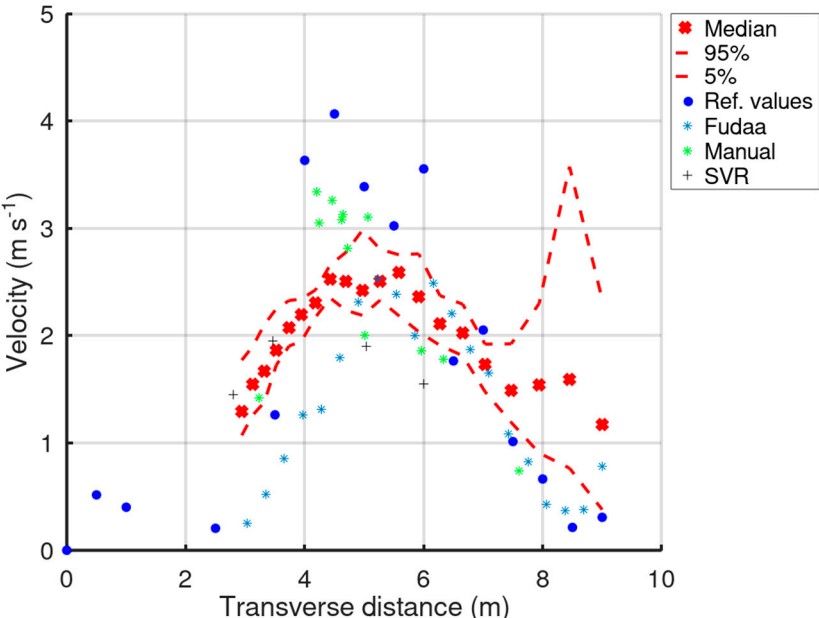

**Figure 12.** Loussios River upstream Atsicholos Bridge. Free-LSPIV 90% confidence interval and median values, the reference values, the surface velocities obtained with Fudaa-LSPIV, the surface velocities estimated manually, and the SVR measurements.

It is worth noting that the narrow confidence interval indicates that Free-LSPIV "feels" quite confident about its results, yet the reference values suggest that the estimated surface velocities have a significant error. However, this "confidence" of Free-LSPIV might not be unrealistic. The surface velocities obtained using a handheld surface velocity radar (SVR) Viatronics SVR-3 Pro ranged between 1.5 and 2.0 m s$^{-1}$. This suggests that the image velocimetry methods correctly capture the velocity of the various surface features. Nevertheless, the problem seems to be that the velocities of these features are not representative of the true surface velocity.

## 4. Discussion

The aforementioned case studies exhibit several characteristics that are common in many image velocimetry applications. The videos of the first three case studies (Kolubara River, Murg River, and Salmon River) were recorded from a UAV employing a high-resolution camera. The remaining two cases were recorded from a medium resolution camera mounted on a tripod. In the first three, the camera recorded the view of the flow from above, whereas in the last two the camera recorded the view from the side (the bank). The orthorectification in the first three cases only requires scaling, whereas in the last two requires projective transformation. The flow velocity is very low (0.14 m s$^{-1}$) at Kolubara River, midrange at Murg River (0.80 m s$^{-1}$) and at Salmon River (0.58 m s$^{-1}$), relatively high (1.76 m s$^{-1}$) at Loussios Atsicholos Bridge, and high (4.07 m s$^{-1}$) with markedly non-uniform flow at Loussios upstream Atsicholos Bridge. The seeding was artificial and fine with high density at Kolubara River, artificial and coarse with high density at Murg River, surface water ripples with low density at Salmon River, floating natural features (leaves, debris) with low density at Loussios at Atsicholos Bridge, and rough water surface at Loussios River upstream Atsicholos Bridge. The conclusions from these case studies are summarized in Table 1.

**Table 1.** Evaluation of LSPIV accuracy under various conditions. Case study 1: Kolubara River, 2: Murg River, 3: Salmon River, 4: Loussios River at Atsicholos Bridge, 5: Loussios River upstream Atsicholos Bridge.

| Case Study | Feature Density | Feature Type | Flow Velocity | Evaluation |
|---|---|---|---|---|
| 1 | High | Artificial and fine | Low | Briefly: relatively good accuracy, medium uncertainty *. The accuracy is relatively good but is compromised by the orthorectification uncertainty which, though low in absolute terms, is significant in relative terms because of the very low flow velocity. The latter combined with the fine (instead of coarse) artificial seeding seem to be the reasons for the spikes of the upper bound of the confidence interval. |
| 2 | High | Artificial and coarse | Midrange | Briefly: very good accuracy, low uncertainty. This is the case study with the best accuracy. The flow velocity is medium and uniform, and the seeding is dense and very distinguishable. The results of the two image velocimetry algorithms practically coincide. Some of the manual estimations lay outside the confidence interval. This is not alarming; it is due to the increased variance of the instantaneous measurements, as the ADCP observations manifest, and the narrowness of the confidence interval. |
| 3 | Medium (varies) | Ripples | Midrange | Briefly: relatively good accuracy, uncertainty varies. In this case study, the accuracy is good at the middle of the cross-section, where there is good agreement between the ADV observations, the two image velocimetry methods, and the manually estimated velocities. The accuracy is low close to the banks because of the lack of detectable features on the surface. This results in the widening of the confidence interval at these locations. |

| Case Study | Feature Density | Feature Type | Flow Velocity | Evaluation |
|---|---|---|---|---|
| 4 | Low | Floating debris | Midrange to high | Briefly: relatively good accuracy, medium uncertainty. In this case study, the accuracy of the image velocimetry methods is relatively good along the entire cross-section. The observed values, the estimated velocities via the image velocimetry methods, and the manual estimations all fall well within the confidence interval. |
| 5 | High | Foam, jet oscillation, roller | High | Briefly: poor accuracy, low uncertainty (deceptive a priori estimation). This case study is interesting as far as the interpretation of the image velocimetry results in case studies with similar conditions is concerned. The confidence interval is narrow and hardly includes any reference value. The image velocimetry methods underestimate the high velocities by 40%. Yet, it may be that the image velocimetry methods correctly estimate the velocity of the surface features, but the velocity of the type of features that are abundant (water foam, roller, or jet oscillation of the hydraulic jump) is not representative of the surface flow velocity. This assumption is also supported by the readings of the hand-held SVR, which are similar with the image velocimetry estimations. A possible explanation could be either the Stokes-drift effect [23] or the effect of the roughness due to the turbulence [32]. On the other hand, the error of the manual estimation, which was based exclusively on within-the-flow features (floating debris and leaves rarely occurring in the footage), was only 20%. |

* 90% confidence intervals obtained by Monte Carlo simulations with Free-LSPIV.

Though in this study we only used PIV-based tools, we argue that other image velocimetry tools, such as PTV, will exhibit similar behavior, since they will also be influenced by the Stokes-drift effect. However, the averaging effect, inherent in PIV due to the cross-correlation of the interrogation windows, imposes limits on the maximum velocity gradients that can be measured [33]. On the other hand, PTV is not subject to the averaging effect, since individual particles are tracked [34]. For this reason, the behavior of PTV in similar conditions deserves investigation.

## 5. Conclusions

In this study, two LSPIV tools were evaluated under five case studies. The first three case studies correspond to situations in which the field survey is performed by experts, whereas the remaining two correspond to conditions where videos are recorded by citizens. The reason for employing two tools was to isolate the peculiarities of the individual tools. This was assured because the velocities estimated using the two tools were similar in all five case studies. Furthermore, surface velocities were manually estimated from the videos in the case studies. The accuracy of the LSPIV tools was relatively good in three out of five case studies, very good at one case study, and poor for the remaining one. In this case study with poor accuracy, typical of conditions met in video footage of extreme flood events obtained by citizens, the surface velocities estimated manually were more accurate than the estimations of the LSPIV tools.

The lessons obtained from this study are that (i) it is always good practice to verify the surface velocities obtained from an image velocimetry method against a few velocities obtained manually; (ii) in the cases where the flow velocity is high and non-uniform, surface velocities obtained with image velocimetry may not be suitable for estimating the average flow velocity of the cross-section; (iii) in such cases, the manually estimated surface velocities may be a more accurate option since an expert would avoid targeting features irrelevant to the flow (e.g., a roller of hydraulic jumps).

Non-uniform hydraulic conditions are, normally, avoided in hydrometry. However, footage obtained from citizens frequently displays non-uniform conditions (attractive to non-experts because of the spectacular phenomenon). Since this footage of extreme flood

events is in many cases the only available source of information, and taking into account the impact these events have on society, it is important to have a very good understanding of how to interpret this information and estimate the involved uncertainty.

**Author Contributions:** Conceptualization, E.R.; methodology, E.R.; software, E.R.; validation, E.R., S.L. and K.M.; formal analysis, E.R.; investigation, E.R.; resources, K.M.; data curation, E.R., S.L. and K.M.; writing—original draft preparation, E.R.; writing—review and editing, S.L. and K.M.; visualization, E.R.; supervision, E.R.; project administration, E.R.; funding acquisition, E.R. and K.M. All authors have read and agreed to the published version of the manuscript.

**Funding:** This research was funded by the Internal Grant/Award of National Observatory of Athens "Low computational burden flood modelling in small to medium-sized water basins in Greece".

**Institutional Review Board Statement:** Not applicable.

**Informed Consent Statement:** Not applicable.

**Data Availability Statement:** The data and tools (Free-LSPIV) used in this study can be found at https://www.hydroshare.org/resource/e713f959ad564adf8acacf1687250ea0/, accessed on 19 April 2022).

**Conflicts of Interest:** The authors declare no conflict of interest. The funders had no role in the design of the study; in the collection, analyses, or interpretation of data; in the writing of the manuscript, or in the decision to publish the results.

## Appendix A

The values of the Fudaa-LSPIV and Free-LSPIV parameters, used in the case studies, are displayed in Tables A1 and A2. The Free-LSPIV parameters that are involved in Monte Carlo simulations are given as a range of values.

**Table A1.** Fudaa-LSPIV parameters of case studies, (I) Kolubara River, (II) Murg River, (III) Salmon River, (IV) Loussios river at of Atsicholos Bridge, (V) Loussios river upstream of Atsicholos Bridge.

|  | I | II | III | IV | V |
|---|---|---|---|---|---|
| Area size (px) | 24 | 40 | 20 | 10 | 16 |
| S1 (px) | 8 | 12 | 6 | 4 | 6 |
| S2 (px) | 8 | 12 | 6 | 4 | 6 |
| S3 (px) | 8 | 12 | 20 | 4 | 6 |
| S4 (px) | 19 | 12 | 6 | 8 | 12 |
| Velocity threshold min (m s$^{-1}$) | 0.02 | 0.0 | 0.0 | 0.20 | 0.1 |
| Velocity threshold max (m s$^{-1}$) | 0.5 | 4.0 | 3.0 | 10.0 | 10.0 |
| Vx threshold min (m s$^{-1}$) | 0.01 | −1.0 | −3.0 | 0.10 | 0.1 |
| Vx threshold max (m s$^{-1}$) | 1.0 | 1.0 | −0.2 | 10.0 | 10.0 |
| Vy threshold min (m s$^{-1}$) | −0.1 | −0.1 | −1.0 | −1.5 | −2.0 |
| Vy threshold max (m s$^{-1}$) | 0.1 | 4.0 | 1.0 | 1.5 | 2.0 |
| Correlation min | 0.6 | 0.4 | 0.4 | 0.40 | 0.30 |
| Correlation max | 0.98 | 0.98 | 0.98 | 0.98 | 0.98 |

**Table A2.** Free-LSPIV parameters of case studies, (I) Kolubara River, (II) Murg River, (III) Salmon River, (IV) Loussios river at of Atsicholos Bridge, (V) Loussios river upstream of Atsicholos Bridge.

|  | I | II | III | IV | V |
|---|---|---|---|---|---|
| IA size (px × px) | 12 × 12–48 × 48 | 30 × 30–120 × 120 | 30 × 30–120 × 120 | 11 × 11–44 × 44 | 10 × 13–40 × 52 |
| SA size (px × px) | 25 × 25–98 × 98 | 51 × 51–206 × 206 | 55 × 55–220 × 220 | 15 × 18–60 × 72 | 32 × 47–130 × 188 |
| Velocity threshold max (m s$^{-1}$) | 0.5 | 4.0 | 3.0 | 3.0 | 10.0 |
| Vx threshold min (m s$^{-1}$) | 0.01 | −1.0 | −3.0 | −1.5 | 0.1 |
| Vx threshold max (m s$^{-1}$) | 1.0 | 1.0 | −0.0 | 3.0 | 10.0 |
| Vy threshold min (m s$^{-1}$) | −0.1 | −4.0 * | −1.0 | −1.5 | −2.0 |
| Vy threshold max (m s$^{-1}$) | 0.1 | 0.1 | 1.0 | 1.5 | 2.0 |
| Correlation min | 0.5–1.0 | 0.5–1.0 | 0.5–1.0 | 0.5–1.0 | 0.5–1.0 |
| Contrast | 0.3–1.0 | 0.3–1.0 | 0.3–1.0 | 0.3–1.0 | 0.3–1.0 |

* In Fudaa-LSPIV *y*-axis points upwards, in Free-LSPIV *y*-axis points downwards.

**Appendix B**

Tables A3 and A4 display the GCP of the Loussios survey upstream and at Atsicholos Bridge. The (row, col) gets the values (1,1) at the upper left corner of the frame, whereas it gets the values (maxrow, maxcol) at the lower right corner, where maxrow × maxcol is the video resolution. All GCP points were taken on the water surface.

**Table A3.** GCP of Loussios river case study at Atsicholos Bridge.

| X Coordinate (m) | Y Coordinate (m) | Frame Row (px) | Frame Column (px) | Error * (m) |
|---|---|---|---|---|
| −3.29 | 14.33 | 391 | 271 | 0.68 |
| −1.90 | 15.67 | 380 | 393 | 0.17 |
| −0.26 | 15.70 | 374 | 513 | 0.11 |
| −3.03 | 5.06 | 724 | 68 | 0.10 |
| −1.53 | 4.75 | 757 | 368 | 0.22 |
| −0.43 | 4.88 | 808 | 617 | 0.09 |
| −2.32 | 8.70 | 521 | 345 | 0.46 |

* Estimated with Fudaa-LSPIV.

**Table A4.** GCP of Loussios river case study upstream Atsicholos Bridge.

| X Coordinate (m) | Y Coordinate (m) | Frame Row (px) | Frame Column (px) | Error (m) |
|---|---|---|---|---|
| −3.66 | 10.07 | 249 | 230 | 0.02 |
| −2.94 | 11.81 | 223 | 353 | 0.24 |
| −1.49 | 12.30 | 219 | 514 | 0.06 |
| 0.24 | 12.90 | 221 | 697 | 0.07 |
| −2.26 | 3.36 | 662 | 66 | 0.02 |
| −1.42 | 4.12 | 588 | 360 | 0.11 |
| −0.36 | 4.17 | 631 | 636 | 0.02 |
| −0.25 | 2.86 | 866 | 678 | 0.05 |

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
