# Peer review of "On the Accuracy of Particle Image Velocimetry with Citizen Videos—Five Typical Case Studies"

_hydrology, doi:10.3390/hydrology9050072_

Round 1

Reviewer 1 Report

General comments:

  1. Paper presents the results of five cases where surface velocities in natural small-sized rivers are recorded using different video techniques, three were done and published by other authors. All five cases were processed using the two LSPIV methods, and in addition, with the manual method and whatever other method was used as reference measurement (ADCP, current meters, or ADV). The methods used and obtained results were clearly presented.
  2. The main question is about the true goal of the paper. The title suggests that the question is asked about the reliability of image velocimetry in different flow conditions (very legitimate issue), while second paragraph in the page two (line 54) suggests that the usage of crowdsourced data might be the true subject? The first sentence of the third paragraph at page two (line 60) does not helps the reader, since it says that paper “address the previous concerns”. So, clear definition of the main papers goal is needed. And that goal has to be supported with an adequate title and the further text. If the usage of the citizen records as the source for image velocimetry is important for the paper, then explanation of cases (2.5.1-2.5.5) must also include the note that the first three were obtained by “professionals” and last two are used as simulated “citizen” recorded cases. Same has to be included in the Discussion section, and in the Conclusions section (last sentence of the first paragraph mentions the citizens footages, line 399, and line 410 too).
  3. The paper is about surface velocity measurement, not the discharge measurement. Please add the clear statement somewhere at the Introduction. I suggest to repeat this also at the Conclusion. We don’t know anything about the cross section, about depths etc. If the “citizen records” of floods are important subject for the paper, then you can elaborate that: after the flood event, the cross section has someone to record, and footages are to be used to assess the depths. So, please pe careful in the text where discharge/flow is mentioned (for example, page 13, line 405).

Details:

Line 7: “…image velocimetry to measure streamflow velocities…” – add “surface” before “streamflow”.

Line 9/10: not only uniform flow, but also developed turbulent velocity profile.

Line 15: You can cut the long sentence: “…including conditions resembling citizen videos, and we conclude…” at the “video.” word.

Line 16: you have five times the “of” word. Rephrase the sentence.

Line 17: “…with non-uniform flow conditions...” – What about extrapolation from the surface velocity to the mean flow velocity? That is linked with the third general comment.

Line 18: “…that the flow velocities…” – use “surface” instead “flow”.

Line 54: “crowdsource” – also the citizen science is used, and you are using citizen footage or similar throughout the text.

Line 59: Fudaa-LSPIV and Free-LSPIV are firstly mentioned here, so give the references.

Line 61: same is for EDF, give the full name and refence.

Lines 104-123: Explain parameters as the difference from previously mentioned parameters in Fudaa-LSPIV. Give explanations what is different and why.

Lines 152, 154, 158: Not the same features are used in radar and LSPIV methods. Radar uses the reflections from the water surface itself, from the small water waves. The question is whether the leaves, wooden mulch/chips, etc. interfere with the radar reflections from the water.

Author Response

Please see the attached PDF file.

Reviewer 2 Report

The manuscript entitled "Is image velocimetry reliable under any condition?" presented five case studies by employing two large-scale particle image velocimetry tools to measure surface velocities and calculate the uncertainty. The results are compared with the manual measurement and reference values. To this end, the authors analysed the performance of two tools under different hydrological conditions. This topic is interesting and the study involves much experiment work. Nevertheless, the manuscript presents some unclear or incomplete scientific reasoning. Some questions about the measurement design, description of uncertainty, and why these five sites were selected should be addressed more straightforwardly. Besides I feel the study would be more valuable if the criteria for evaluating the performance of image velocimetry tools were clear and the recommendations on how to increase the reliability accordingly to different conditions could be included. Therefore, my recommendation is major revision before reconsideration for publication in Hydrology. Following are some of the issues which need authors' attention。

Author Response

Please see the attached PDF file.

Reviewer 3 Report

The paper shows performances of particle image velocimetry in five case studies, corresponding to conditions more or less suitable for particle hydrometry. Authors employed two different tools claiming to obtain more general conclusions regarding “all image velocimetry methods” and  estimated the surface velocities from videos also by supervised procedure.

The paper does not contain original contributions and just describes the results obtained analyzing inhomogeneous case studies with available PIV tools. It seems also unclear the Author’s statement about different “flavors” of analysis techniques confusing PIV and PTV, each one with specific pros and cons.

The introduction seems inadequate to provide a solid background of both the paper aims and methods. Inexplicably, Authors used a supervised analysis technique very similar to a published PTV analysis technique without referring to the original paper. Moreover, the attained results are similar to those presented in the literature but no reference to those works has been reported. Other methods are available in literature (like OTV) and should be cited. Accordingly, I suggest to widen the introduction section.

The title should be changed since it does not reflect the content of the paper and should be less ambitious.

The results obtained in the case studies are unsatisfactorily described, without necessary details on the specific monitoring condition (seeding density, characteristics and latency, etc.) as well as on accuracy of the (different) used instrumentations, mainly for those used to measure the reference values. Regarding the PIV analysis tools videos extraction should be detailed and seeding influence on the obtained results considered.

The benchmark values for the image velocimetry are obtained with a supervised PTV but the values obtained seem non statistically robust inasmuch as they are evaluated for few (one?) tracers. Accordingly, additional details are needed and more robust analysis should be performed over the entire videos. Additionally, figures are someway blurring inasmuch as they present many overlapping results and inhomogeneous data. I suggest to remove river cross-sections (eventually presenting them in section 2) and narrowing the vertical axis range, even if the confidence band would be partially drawn.

Finally, I propose to publish the manuscript after a major revision.

Author Response

Please see the attached PDF file.

Round 2

Reviewer 2 Report

This revised manuscript has not been discussed and improved at a deeper level, and it is recommended not to be published.

(Editor's Note: The reviewer added comments by email later, please check the attachment.)

Author Response

We are sorry the reviewer was not satisfied by our improvements. We replied point-by-point to his initial review and did everything possible to address his suggestions within the limits of our available resources.

Reviewer 3 Report

The paper has been improved according to the comments and can be published after an accurate English style and typos revision

Author Response

We appreciate the constructive comments of the reviewer, and we are happy she/he is satisfied with our improvements.